# A Statistical Framework for Low-bitwidth Training of Deep Neural Networks

**Jianfei Chen, Yu Gai, Zhewei Yao, Michael W. Mahoney, and Joseph E. Gonzalez**
University of California, Berkeley
{jianfeic, yu_gai, zheweiy, mahoneymw, jegonzal}@berkeley.edu

## Abstract

Fully quantized training (FQT), which uses low-bitwidth hardware by quantizing the activations, weights, and gradients of a neural network model, is a promising approach to accelerate the training of deep neural networks. One major challenge with FQT is the lack of theoretical understanding, in particular of how gradient quantization impacts convergence properties. In this paper, we address this problem by presenting a statistical framework for analyzing FQT algorithms. We view the quantized gradient of FQT as a stochastic estimator of its full precision counterpart, a procedure known as quantization-aware training (QAT). We show that the FQT gradient is an unbiased estimator of the QAT gradient, and we discuss the impact of gradient quantization on its variance. Inspired by these theoretical results, we develop two novel gradient quantizers, and we show that these have smaller variance than the existing per-tensor quantizer. For training ResNet-50 on ImageNet, our 5-bit block Householder quantizer achieves only 0.5% validation accuracy loss relative to QAT, comparable to the existing INT8 baseline. Our code is publicly available at `https://github.com/cjf00000/StatQuant`.

## 1 Introduction

Deep neural networks (DNNs) have a high computational cost and memory footprint that slow down their training and inference. By taking advantage of low-bitwidth computational units in hardware, neural network quantization methods provide promising approaches for reducing the cost of timing, memory, and energy consumption, for both training and inference.

Notable quantization methods can be mainly categorized into two groups, inference quantization and training quantization. In *inference quantization*, the weights and the activations are quantized to speed up the inference phase. Among inference quantization approaches, *post training quantization* usually does not require access to the partial/full training dataset, and it does not need to re-train/fine-tune the quantized model [1, 2, 3, 4, 5]. To reduce the performance gap between the quantized model and its full precision counterpart, *quantization-aware training* (QAT) fine-tunes the quantized model on the training dataset [6, 7, 8, 9, 10, 11, 12, 13, 14]. However, QAT computes the gradients in full precision, so the training phase is not accelerated.

*Training quantization* methods, also known as *fully quantized training* (FQT), further quantize the gradients, compared with QAT. In FQT, all the activations, weights, and gradients are quantized in both the forward and backward propagation. Hence, training can be implemented efficiently on low-bitwidth computational units, such as tensor cores [15]. Low-bitwidth hardware is faster and more power-efficient, as compared to FP32 counterparts. As the need for training huge models continues to grow [16, 17, 18], there has been increasing attention on FQT. Earlier work on FQT includes mixed-precision FP16/FP32 training [19] and lossy 2-bit training [6]. Recently, 8-bit FQT has emerged as a sweet spot on the accuracy versus efficiency tradeoff. Various 8-bit numerical formats have been proposed, including INT8 [20, 21, 22, 23], FP8 [24, 25], block floating point [26], FP8

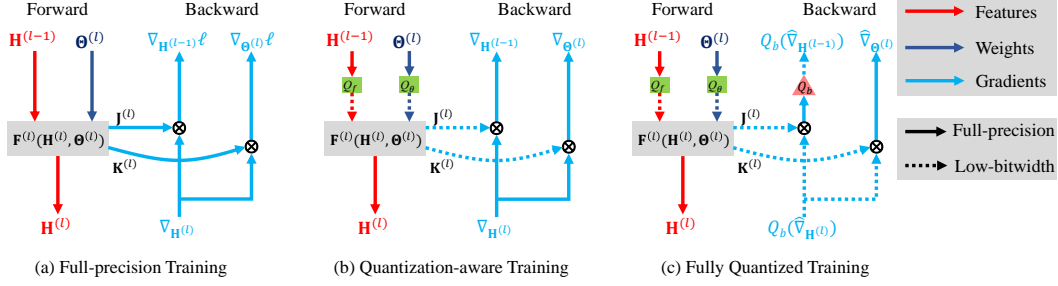

Figure 1: Computational graphs for full-precision and quantized training settings.

with learnable parameters [27], and adaptive precision [28]. Several of them achieved near-lossless ($\leq 0.4\%$) validation accuracy for training ResNet-50 [29] on ImageNet [26, 22].

Despite abundant empirical results on FQT, the theoretical understanding is still lacking. Studying the effect of gradient quantization is challenging, due to the error accumulation caused by recursively quantizing the gradient at each layer. Existing theoretical results are based on very strong assumptions, such as untrained single layer networks [20] or convex objective functions [22]. To the best of our knowledge, there is not yet a bound on how the quantization scheme (bitwidth, type of quantizer) affects the quality of the quantized gradient.

In this paper, we present a general framework for FQT algorithms with theoretical guarantees. Unlike existing work [20, 22], which studies the *worst case* behavior of the gradient, we adopt a statistical approach. The FQT gradient can be viewed as a *stochastic estimator* of the QAT gradient, and we analyze the quantized gradient through its bias and variance. We provide theoretical bounds to guide practice, and we show how to use these theoretical results to lead to improved performance in practice. Our framework makes minimal assumption: deterministic forward propagation and unbiased stochastic gradient quantizer. Our main contributions include the following.

1. We present a framework for FQT and use the framework to show that the FQT gradient is an *unbiased* estimator of the QAT gradient. This implies that FQT and QAT algorithms eventually have the same convergence behavior, when the learning rate goes to zero.
2. We provide a general formula for the variance of the FQT gradient, and discuss the impact of bitwidth on gradient variance for the per-tensor gradient quantizer in existing FQT algorithms.
3. We propose two novel gradient quantizers for FQT, which significantly reduce variance. Our quantizers address the large dynamic range variation across gradient samples and spread the signal across the gradient dimensions.
4. We evaluate our quantizers on ImageNet using ResNet50 and reduce the gradient encoding from 8-bits to 5-bits without loss in validation accuracy.

## 2 Framework for Fully Quantized Training

In this section, we describe the mathematical formulation and assumptions of our framework. Throughout this paper, we use uppercase and lowercase letters ($\mathbf{A}$/$\mathbf{b}$) to denote matrices and row vectors, respectively. The $i$-th row and the $j$-th column of matrix $\mathbf{A}$ are denoted as $\mathbf{a}_i$ and $\mathbf{A}_{:,j}$, respectively. The operator $\text{vec}(\mathbf{A})$ stands for reshaping $\mathbf{A}$ into a row vector. For a matrix $\mathbf{A}$, $\|\mathbf{A}\|_F^2$ is the Frobenius norm and $\|\mathbf{A}\|_2^2$ is the $L_2$ operator norm. Furthermore, $\mathbf{e}_i$ is the $i$-th indicator vector; $\mathbf{1}$ is the all-one vector; and $[N] = \{0, 1, \ldots, N\}$, $[N]_+ = \{1, 2, \ldots, N\}$ are sets of integers. A table of notations can be found in Appendix A.

We assume that the DNN model $\mathbf{F}(\cdot; \boldsymbol{\Theta})$ is composed of $L$ layers with the learnable parameter $\boldsymbol{\Theta}$. The forward propagation is

$$\mathbf{H}^{(0)} = \mathbf{X}, \quad \mathbf{H}^{(l)} = \mathbf{F}^{(l)}\left(\mathbf{H}^{(l-1)}; \boldsymbol{\Theta}^{(l)}\right), \quad \mathbf{F}(\mathbf{X}; \boldsymbol{\Theta}) = \mathbf{H}^{(L)}, \quad \forall l \in [L]_+, \tag{1}$$

where $\mathbf{X} \in \mathbb{R}^{N \times D}$ is a batch of data ($N$ is the batch size, and $D$ is the feature dimensionality), and $\mathbf{F}(\mathbf{X}; \boldsymbol{\Theta}) \in \mathbb{R}^{N \times C}$ is the prediction ($C$ is the number of labels). Here, $\mathbf{F}^{(l)}$ is the $l$-th layer of the model with parameter $\boldsymbol{\Theta}^{(l)}$, and $\mathbf{H}^{(l)}$ is an $N \times D^{(l)}$-dimensional feature map (a.k.a. activations)

after the $l$-th layer. To optimize the parameter $\mathbf{\Theta}$, the following empirical risk is minimized,

$$\min_{\mathbf{\Theta}} \mathcal{L}(\mathbf{\Theta}) := \mathbb{E}\left[\ell\left(\mathbf{F}(\mathbf{X}; \mathbf{\Theta}), \mathbf{Y}\right)\right], \tag{2}$$

where $\mathbf{Y} \in \mathbb{R}^{N \times C}$ is the corresponding label of $\mathbf{X}$, $\ell$ is the loss function on a batch of labels, and the expectation is taken over all possible batches from a training dataset. Stochastic gradient descent (SGD) [30] is oftentimes used to solve the above problem, which takes the update $\mathbf{\Theta}_{t+1} = \mathbf{\Theta}_t - \eta_t \nabla_{\mathbf{\Theta}_t} \ell\left(\mathbf{F}(\mathbf{X}; \mathbf{\Theta}_t), \mathbf{Y}\right)$, where $\eta_t$ is the $t$-th step learning rate.

## 2.1 Quantization Aware Training

To accelerate inference, the forward propagation Eq. (1) is quantized as follows,

$$\forall l \in [L]_+, \quad \tilde{\mathbf{H}}^{(l-1)} = Q_f\left(\mathbf{H}^{(l-1)}\right), \quad \tilde{\mathbf{\Theta}}^{(l)} = Q_\theta\left(\mathbf{\Theta}^{(l)}\right), \quad \mathbf{H}^{(l)} = \mathbf{F}^{(l)}\left(\tilde{\mathbf{H}}^{(l-1)}; \tilde{\mathbf{\Theta}}^{(l)}\right), \tag{3}$$

where $Q_f(\cdot)$ and $Q_\theta(\cdot)$ are quantizers for features and weights, and $\tilde{\mathbf{H}}^{(l-1)}$ and $\tilde{\mathbf{\Theta}}^{(l)}$ are the quantized versions of $\mathbf{H}^{(l-1)}$ and $\mathbf{\Theta}^{(l)}$. For the particular case of linear layers, e.g., fully-connected and convolutional layers, forward propagation can be written as $\mathbf{F}^{(l)}\left(\tilde{\mathbf{H}}^{(l-1)}; \tilde{\mathbf{\Theta}}^{(l)}\right) = \tilde{\mathbf{H}}^{(l-1)} \tilde{\mathbf{\Theta}}^{(l)}$, and can be implemented efficiently with low-bitwidth computing kernels [31, 32]. Our framework assumes that the entire forward propagation is *deterministic*. That is to say, the quantizers $Q_f(\cdot)$ and $Q_\theta(\cdot)$ must be deterministic, and stochastic layers such as dropout are not allowed. This assumption aligns with current state-of-the-art inference quantization approaches [6, 7, 8, 2, 3, 4, 5].

QAT trains the quantized model (Eq. 3) on a training dataset. Incorporating the chain rule and using the straight-through estimator (STE) [33] of quantizers, which assumes that the gradient directly flow though the non-differentiable quantizer, QAT defines the gradient with back-propagation as:

$$\forall l \in [L]_+, \quad \mathrm{vec}(\nabla_{\mathbf{\Theta}^{(l)}}) := \mathrm{vec}(\nabla_{\mathbf{H}^{(l)}})\mathbf{K}^{(l)}, \quad \mathrm{vec}(\nabla_{\mathbf{H}^{(l-1)}}) := \mathrm{vec}(\nabla_{\mathbf{H}^{(l)}})\mathbf{J}^{(l)}, \tag{4}$$

where $\mathbf{J}^{(l)} := \frac{\partial \mathrm{vec}(\mathbf{H}^{(l)})}{\partial \mathrm{vec}(\tilde{\mathbf{H}}^{(l-1)})}$, $\mathbf{K}^{(l)} := \frac{\partial \mathrm{vec}(\mathbf{H}^{(l)})}{\partial \mathrm{vec}(\tilde{\mathbf{\Theta}}^{(l)})}$ are two Jacobian matrices. We refer to $\nabla_{\mathbf{\Theta}^{(l)}}, \nabla_{\mathbf{H}^{(l-1)}}$ as the QAT gradient , which provides approximate descent directions of the discrete learning problem (2), and we denote $\nabla_{\mathbf{\Theta}} = \{\nabla_{\mathbf{\Theta}^{(l)}}\}$. The shape of $\nabla_{\mathbf{\Theta}^{(l)}}, \nabla_{\mathbf{H}^{(l-1)}}$ is the same with corresponding parameter $\mathbf{\Theta}^{(l)}$ and feature $\mathbf{H}^{(l-1)}$. For linear layers, backpropagation can be written as $\nabla_{\mathbf{\Theta}^{(l)}} = \tilde{\mathbf{H}}^{(l-1)^\top} \nabla_{\mathbf{H}^{(l)}}$ and $\nabla_{\mathbf{H}^{(l-1)}} = \nabla_{\mathbf{H}^{(l)}} \tilde{\mathbf{\Theta}}^{(l)^\top}$. Since $\nabla_{\mathbf{H}^{(l)}}$ is not quantized, the back propagation in QAT cannot be implemented with low-bitwidth kernels.

## 2.2 Fully Quantized Training

To make back propagation more efficient, FQT further quantizes the gradients at each layer as:

$$\forall l \in [L]_+, \quad \mathrm{vec}(\hat{\nabla}_{\mathbf{\Theta}^{(l)}}) := \mathrm{vec}(Q_b(\hat{\nabla}_{\mathbf{H}^{(l)}}))\mathbf{K}^{(l)}, \quad \mathrm{vec}(\hat{\nabla}_{\mathbf{H}^{(l-1)}}) := \mathrm{vec}(Q_b(\hat{\nabla}_{\mathbf{H}^{(l)}}))\mathbf{J}^{(l)}, \tag{5}$$

where $\hat{\nabla}_{\mathbf{H}^{(L)}} := \nabla_{\mathbf{H}^{(L)}}$, and $Q_b(\cdot)$ is an *unbiased stochastic* quantizer, i.e., $\mathbb{E}\left[Q_b(\mathbf{X})\right] = \mathbf{X}$, for any $\mathbf{X}$. Such stochastic quantizers are typically implemented with stochastic rounding [34], and they are already widely adopted in existing FQT approaches [20, 22, 26]. We refer to $\hat{\nabla}_{\mathbf{\Theta}^{(l)}}, \hat{\nabla}_{\mathbf{H}^{(l-1)}}$ as the FQT gradient, and we denote $\hat{\nabla}_{\mathbf{\Theta}} = \{\hat{\nabla}_{\mathbf{\Theta}^{(l)}}\}$. For linear layers, back propagation reduces to

$$\hat{\nabla}_{\mathbf{\Theta}^{(l)}} = \tilde{\mathbf{H}}^{(l-1)^\top} Q_b(\hat{\nabla}_{\mathbf{H}^{(l)}}), \quad \hat{\nabla}_{\mathbf{H}^{(l-1)}} = Q_b(\hat{\nabla}_{\mathbf{H}^{(l)}}) \tilde{\mathbf{\Theta}}^{(l)^\top}, \tag{6}$$

which can be implemented with low-bitwidth kernels since both operands are now quantized.

The relationship between full-precision training, QAT, and FQT is illustrated in Fig. 1. Full-precision training and QAT solve different learning problems, since full-precision training optimizes the exact model (Eq. 1), while QAT approximately optimizes the quantized model (Eq. 3). In contrast, QAT and FQT aim to optimize the same model, but with different gradient estimators: QAT uses $\nabla_{\mathbf{\Theta}}$ and FQT uses $\hat{\nabla}_{\mathbf{\Theta}}$. In this paper, we study the difference between FQT and QAT by comparing these gradients. On the other hand, improving QAT towards full-precision training, which typically involves designing a better network $\mathbf{F}(\cdot; \mathbf{\Theta})$ and learnable quantizers, is a different problem outside the scope of this paper. We refer readers to [8, 9, 12] for state-of-the-art approaches for QAT, which can be potentially combined with this paper to reduce the bitwidth of the forward propagation.

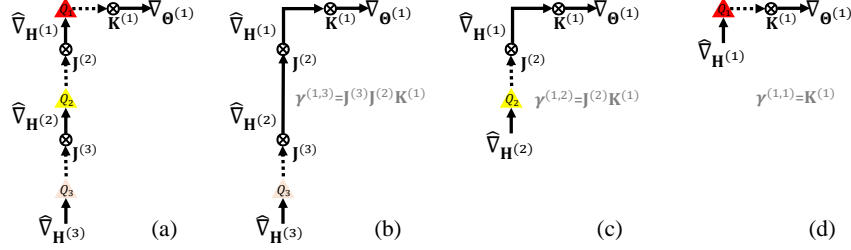

Figure 2: Decomposition of the variance. (a) Computational subgraph for the first layer parameter gradient $\hat{\nabla}_{\Theta^{(1)}}$; (b) $\text{vec}(Q_b(\hat{\nabla}_{\mathbf{H}^{(3)}}))\boldsymbol{\gamma}^{(1,3)}$; (c) $\text{vec}(Q_b(\hat{\nabla}_{\mathbf{H}^{(2)}}))\boldsymbol{\gamma}^{(1,2)}$; (d) $\text{vec}(Q_b(\hat{\nabla}_{\mathbf{H}^{(1)}}))\boldsymbol{\gamma}^{(1,1)}$.

## 3 Theoretical Results

We view the FQT gradient $\hat{\nabla}_{\Theta}$ as a stochastic estimator of the QAT gradient $\nabla_{\Theta}$. The FQT gradient $\hat{\nabla}_{\Theta}$ has $L+1$ sources of randomness. The first one is brought by randomly subsampling the batch $\mathcal{B} = (\mathbf{X}, \mathbf{Y})$, and it is shared with the QAT gradient. The other $L$ sources of randomness are due to the stochastic quantizers $Q_b(\cdot)$ per each layer, as illustrated in Fig. 2(a).

Both QAT and FQT can be viewed as stochastic optimization algorithms to solve the learning problem (2) approximately. We can analyze the behavior of these algorithms through the bias and variance of the gradient. All the proofs in this section can be found in Appendix C.

### 3.1 Bias

The following theorem states that the FQT gradient is an unbiased estimator of the QAT gradient.

**Theorem 1.** *(Unbiased gradient) The FQT gradient $\hat{\nabla}_{\Theta}$ defined as Eq. (5) is an unbiased estimator of the QAT gradient defined as Eq. (4), i.e.,* $\mathbb{E}\left[\hat{\nabla}_{\Theta} \mid \mathcal{B}\right] = \nabla_{\Theta}$.

In standard SGD theory [35], an unbiased gradient implies convergence to a stationary point. QAT and FQT can be viewed as SGD algorithms that approximately solve the learning problem (2). More rigorously, we can view QAT and FQT as stochastic approximation [30] algorithms for finding the root $\mathbb{E}\left[\nabla_{\Theta}\right] = 0$, where QAT updates as $\Theta_{t+1} = \Theta_t - \eta_t \nabla_{\Theta_t}$, and FQT has a more noisy update $\Theta_{t+1} = \Theta_t - \eta_t \hat{\nabla}_{\Theta_t}$. Intuitively, when the step size $\eta_t \to 0$, both algorithms simulate the ordinary differential equation $\frac{d\Theta}{dt} = -\mathbb{E}\left[\nabla_{\Theta}\right]$ (Theorem 2.1 in [36]). Therefore, QAT and FQT are equivalent at the continuous limit, regardless of the choice of the specific gradient quantizer $Q_b(\cdot)$.

### 3.2 Variance

We define the variance of a random matrix $\mathbf{X}$ as the summation of the variance for each entry, i.e., $\text{Var}\left[\mathbf{X}\right] := \sum_i \text{Var}\left[\text{vec}(\mathbf{X})_i\right] = \mathbb{E}\left\|\text{vec}(\mathbf{X}) - \mathbb{E}\left[\text{vec}(\mathbf{X})\right]\right\|_2^2 = \mathbb{E}\left\|\mathbf{X} - \mathbb{E}\left[\mathbf{X}\right]\right\|_F^2$.

The convergence rate of stochastic optimization algorithms depends on the *variance*. For example, for SGD with nonconvex and smooth objective functions, the convergence rate of the gradient norm $\mathbb{E}\left\|\nabla_{\Theta_t}\right\|^2$ is $O(\sigma/\sqrt{T})$ w.r.t. the numbers of total iterations $T$ (Corollary 2.2 in [37]), if the gradient variance is bounded by $\sigma^2$. Therefore, larger variance leads to more iterations to converge. The variance of the FQT gradient is given by the following theorem.

**Theorem 2.** *(Gradient Variance) For all $k \le l$, let* $\boldsymbol{\gamma}^{(k,l)} = \left(\prod_{i=l}^{k+1} \mathbf{J}^{(i)}\right) \mathbf{K}^{(k)}$*, then*

$$\text{Var}\left[\hat{\nabla}_{\Theta}\right] = \text{Var}\left[\nabla_{\Theta}\right] + \sum_{l=1}^{L} \mathbb{E}\left[\sum_{k=1}^{l} \text{Var}\left[\text{vec}(Q_b(\hat{\nabla}_{\mathbf{H}^{(l)}}))\boldsymbol{\gamma}^{(k,l)} \mid \hat{\nabla}_{\mathbf{H}^{(l)}}\right]\right] \tag{7}$$

$$\le \text{Var}\left[\nabla_{\Theta}\right] + \sum_{l=1}^{L} \mathbb{E}\left[\text{Var}\left[Q_b(\hat{\nabla}_{\mathbf{H}^{(l)}}) \mid \hat{\nabla}_{\mathbf{H}^{(l)}}\right] \sum_{k=1}^{l} \left\|\boldsymbol{\gamma}^{(k,l)}\right\|_2^2\right]. \tag{8}$$

Intuitively, Eq. (7) decomposes the variance in the FQT gradient into each of its $L+1$ sources of randomnesses. Particularly, the first term $\text{Var}\left[\nabla_{\Theta}\right]$ is the variance of the QAT gradient, which

considers the variance from subsampling the batch $\mathcal{B}$. All the remaining variance terms come from gradient quantization, where each term $\mathrm{Var}\left[\mathrm{vec}(Q_b(\hat{\nabla}_{\mathbf{H}^{(l)}}))\boldsymbol{\gamma}^{(k,l)} \mid \hat{\nabla}_{\mathbf{H}^{(l)}}\right]$ is the variance imposed by the quantizer $Q_b(\hat{\nabla}_{\mathbf{H}^{(l)}})$ on layer $l$ to the gradient $\hat{\nabla}_{\boldsymbol{\Theta}^{(k)}}$ on layer $k$. For example, in a 3-layer network, consider the variance of the first layer parameter gradient $\hat{\nabla}_{\boldsymbol{\Theta}^{(1)}}$, whose computational subgraph is illustrated in Fig. 2(a). The *gradient variance* $\mathrm{Var}\left[\hat{\nabla}_{\boldsymbol{\Theta}^{(1)}} \mid \mathcal{B}\right]$ is challenging to analyze since $\hat{\nabla}_{\boldsymbol{\Theta}^{(1)}}$ is affected by all the three quantizers $Q_1, Q_2, Q_3$, which entangle with the computing operations. Eq. (7) disentangles the variance introduced by each quantizer, where the term $\mathrm{vec}(Q_b(\hat{\nabla}_{\mathbf{H}^{(l)}}))\boldsymbol{\gamma}^{(1,l)}$ computes $\boldsymbol{\Theta}^{(1)}$ based on $\hat{\nabla}_{\mathbf{H}^{(l)}}$, as if there is only one quantizer $Q_b(\hat{\nabla}_{\mathbf{H}^{(l)}})$ along the path (Fig. 2(b)-(d)). The variance of $\mathrm{vec}(Q_b(\hat{\nabla}_{\mathbf{H}^{(l)}}))\boldsymbol{\gamma}^{(1,l)}$ is simpler to analyze since it is only a linear transformation of the quantized value. Particularly, we can upper bound the variance with Eq. (8). The bound only depends on the *quantizer variance* $\mathrm{Var}\left[Q_b(\hat{\nabla}_{\mathbf{H}^{(l)}}) \mid \hat{\nabla}_{\mathbf{H}^{(l)}}\right]$, weighted by the deterministic term $\sum_{k=1}^{l}\left\|\boldsymbol{\gamma}^{(k,l)}\right\|_2^2$. Therefore, Theorem 2 reduces the complicated problem on *gradient variance* into the simple problem on *quantizer variance*.

### 3.3 Case Study: FQT with Per-tensor Quantizer

We now analyze the quantizer variance for a specific *per-tensor quantizer* (PTQ), which is widely adopted in existing INT8 training approaches [20, 22]. The quantizer is defined as

$$Q_b(\hat{\nabla}_{\mathbf{H}^{(l)}}) = \mathrm{SR}\left(S^{(l)}(\hat{\nabla}_{\mathbf{H}^{(l)}} - Z^{(l)})\right)/S^{(l)} + Z^{(l)}, \text{ where } \mathrm{SR}(X) = \begin{cases} \lceil X \rceil & \text{with prob. } X - \lfloor X \rfloor \\ \lfloor X \rfloor & \text{otherwise} \end{cases}.$$

If the bitwidth is $b$-bits, the affine transformation $S^{(l)}(\hat{\nabla}_{\mathbf{H}^{(l)}} - Z^{(l)})$ maps the gradient to an interval $[0, B]$ of $B = 2^b - 1$ bins. It is then rounded to integers in $[B]$ by the stochastic rounding [34] operation $\mathrm{SR}(\cdot)$. Finally, the quantized value is mapped back by an inverse transformation, which is often carried out implicitly. We take the zero point $Z^{(l)} = \min \hat{\nabla}_{\mathbf{H}^{(l)}}$ and the scale $S^{(l)} = B/R(\hat{\nabla}_{\mathbf{H}^{(l)}}) = B/R(\mathbf{X})$, where $R(\mathbf{X}) = \max \mathbf{X} - \min \mathbf{X}$ is often referred as the *dynamic range* of $\mathbf{X}$. Since the SR operation is applied independently to each entry, we have the quantizer variance

$$\mathrm{Var}\left[Q_b(\hat{\nabla}_{\mathbf{H}^{(l)}}) \mid \hat{\nabla}_{\mathbf{H}^{(l)}}\right] = \frac{1}{(S^{(l)})^2}\mathrm{Var}\left[\mathrm{SR}(\cdot) \mid \hat{\nabla}_{\mathbf{H}^{(l)}}\right] \leq \frac{ND^{(l)}}{4B^2}R(\hat{\nabla}_{\mathbf{H}^{(l)}})^2, \tag{9}$$

where the variance of stochastic rounding reaches the maximum $1/4$ when the input falls to the center of a bin. Combining Eq. (9) and Eq. (8), we can bound the gradient variance by

$$\mathrm{Var}\left[\hat{\nabla}_{\boldsymbol{\Theta}}\right] \leq \mathrm{Var}\left[\nabla_{\boldsymbol{\Theta}}\right] + \frac{N}{4B^2}\sum_{l=1}^{L}D^{(l)}\mathbb{E}\left[R(\hat{\nabla}_{\mathbf{H}^{(l)}})^2\sum_{k=1}^{l}\left\|\boldsymbol{\gamma}^{(k,l)}\right\|_2^2\right]. \tag{10}$$

This gives us some insight on the impact of gradient bitwidth to the variance. Particularly, when the bitwidth $b$ is high ($B = 2^b - 1$), the second term is negligible compared with the variance of the QAT gradient itself. In this case, we can reduce the gradient bitwidth for free. As $b$ getting smaller, the quantization variance domainates, and each less bit increase the variance by 4x. The rapid increase of variance makes it very challenging for existing INT8 training approaches to work for lower bitwidth.

## 4 Variance Reduced Quantizers

To reduce gradient variance, we propose a new family of gradient quantizers that have smaller variance than existing PTQ. Particularly, we extend PTQ with a scale *matrix* and a zero point *vector*:

$$Q_b(\hat{\nabla}_{\mathbf{H}^{(l)}}) = (\mathbf{S}^{(l)})^{-1}\mathrm{SR}\left(\mathbf{S}^{(l)}(\hat{\nabla}_{\mathbf{H}^{(l)}} - \mathbf{1}\mathbf{z}^{(l)})\right) + \mathbf{1}\mathbf{z}^{(l)}, \tag{11}$$

where $\mathbf{S}^{(l)}$ is a $N \times N$ matrix, $\mathbf{1}$ is a row vector, and $\mathbf{z}^{(l)}$ is a column vector. This quantizer scales and rotates the rows of the gradient, and reduces to the PTQ when $\mathbf{S}^{(l)} = S^{(l)}\mathbf{I}$ and $\mathbf{z}^{(l)} = \mathbf{1}^\top Z^{(l)}$. The variance of this quantizer is $\mathrm{Var}\left[Q_b(\hat{\nabla}_{\mathbf{H}^{(l)}}) \mid \hat{\nabla}_{\mathbf{H}^{(l)}}\right] \leq \frac{D^{(l)}}{4}\left\|(\mathbf{S}^{(l)})^{-1}\right\|_F^2$. This can be minimized as

$$\min_{\mathbf{S}^{(l)}}\left\|(\mathbf{S}^{(l)})^{-1}\right\|_F^2, \text{ s.t. } R(\mathbf{S}^{(l)}\hat{\nabla}_{\mathbf{H}^{(l)}}) \leq B, \tag{12}$$

where the constraint ensures that the inputs are mapped within $[0, B]$. The derivation of variance for all quantizers in this section can be found in Appendix D.

## 4.1 Per-sample Quantizer

We introduce *per-sample quantizer* (PSQ), which addresses the large variation of dynamic range across samples. PSQ takes $\mathbf{S}^{(l)} = \text{diag}\{s_1, \ldots, s_N\}$ as a diagonal matrix. Note that the activation gradient $\hat{\nabla}_{\mathbf{H}^{(l)}}$ is an $N \times D^{(l)}$ matrix, where each row is a sample and each column is a feature. In this case, $\mathbf{S}^{(l)}(\hat{\nabla}_{\mathbf{H}^{(l)}} - \mathbf{1}\mathbf{z}^{(l)})$ can be thought as applying different scale per each sample (row) of the gradient $\hat{\nabla}_{\mathbf{H}^{(l)}}$. PSQ takes $O(ND^{(l)})$ FP32 operations to compute the affine transformation, and it is already available in the FBGEMM library [32]. Solving problem (12) gives the optimal transformation $s_i = B/R(\hat{\nabla}_{\mathbf{h}_i^{(l)}})$, where $R(\hat{\nabla}_{\mathbf{h}_i^{(l)}}) = \max \hat{\nabla}_{\mathbf{h}_i^{(l)}} - \min \hat{\nabla}_{\mathbf{h}_i^{(l)}}$ is the dynamic range of the $i$-th row. Dynamic range is important since it directly affects the quantization bin size, and hence the gradient variance. The variance of PSQ is $\text{Var}\left[Q_b(\hat{\nabla}_{\mathbf{H}^{(l)}}) \mid \hat{\nabla}_{\mathbf{H}^{(l)}}\right] \leq \frac{D^{(l)}}{4B^2}\sum_{i=1}^N R(\hat{\nabla}_{\mathbf{h}_i^{(l)}})^2$. Since $R(\hat{\nabla}_{\mathbf{H}^{(l)}}) = \max_i R(\hat{\nabla}_{\mathbf{h}_i^{(l)}})$, PSQ has smaller variance than PTQ (Eq. 9).

The variance reduction of PSQ relative to PTQ is significant, due to the *sparsity of gradient*. Consider a cross-entropy loss $\ell(\mathbf{H}^{(L)}, \mathbf{Y}) = \sum_{i=1}^N \sum_{c=1}^C Y_{ic} \log \text{Softmax}(\mathbf{h}_i^{(L)})_c$ with the gradient $\hat{\nabla}_{\mathbf{h}_i^{(L)}} = \mathbf{y}_i - \text{Softmax}(\mathbf{h}_i^{(L)})$. If a sample $i$ is correctly classified, $\text{Softmax}(\mathbf{h}_i^{(L)})$ should be close to $\mathbf{y}_i$. Since the *training* accuracy of DNNs is usually near 100%, the dynamic range $R(\hat{\nabla}_{\mathbf{h}_i^{(L)}})$ should be close to zero for most samples except some outliers, as illustrated in Fig. 4. In this case, PTQ is very ineffective since the quantization range is unnecessarily large for correctly classified samples.

## 4.2 Block Householder Quantizer

Quantization treats all samples equally, yet often only a few samples (rows) of the gradient, $\hat{\nabla}_{\mathbf{H}^{(L)}}$, are significant, and the rest waste precious bits to encode zeros. We want a mechanism to spread the "signal" in informative samples across the entire gradient representation. Consider an extreme case when all rows except the first row of $\hat{\nabla}_{\mathbf{H}^{(L)}}$ are close to zero. That is, let $\lambda_1 = R(\hat{\nabla}_{\mathbf{h}_1^{(L)}})$, $\lambda_2 = 2\max_{i \neq 1} \left\|\hat{\nabla}_{\mathbf{h}_i^{(L)}}\right\|_\infty$, and assume $\lambda_2/\lambda_1 \approx 0$. In this case, all the rows other than the first row carry almost no information, since they are too small relative to the first row. However, they consume $(N-1)/N$ of the total computation.

To utilize the wasted bits in the last $N-1$ rows, we construct $\mathbf{S}^{(l)} = \mathbf{Q}\text{diag}(s_1, s_2, \ldots, s_2)$, where $\mathbf{Q} = \mathbf{I} - 2\mathbf{n}\mathbf{n}^\top/\|\mathbf{n}\|_2^2$ is a Householder reflection with the normal vector $\mathbf{n} = 1/\sqrt{N} - \mathbf{e}_1$. Intuitively, $\mathbf{Q}$ maps a coordinate vector $\mathbf{e}_1$ to an all-one vector $\mathbf{1}/\sqrt{N}$, spreading out the large numbers in the first row evenly into other rows. Taking $s_1 \propto \lambda_1^{-1/3} N^{1/6}$ and $s_2 \propto \lambda_2^{-1/3} N^{1/6}$, we have

$$\text{Var}\left[Q_b(\hat{\nabla}_{\mathbf{H}^{(l)}}) \mid \hat{\nabla}_{\mathbf{H}^{(l)}}\right] \leq \frac{D^{(l)}}{4B^2}\left(\lambda_1^{2/3}N^{-1/3} + \lambda_2^{2/3}N^{2/3}\right)^3 \approx \frac{D^{(l)}}{4B^2}\lambda_1^2 N^{-1} = O(\lambda_1^2/N).$$

Additional rows now *reduce* the variance, since they alleviate the burden of the first row. For comparison, PTQ has $O(N\lambda_1^2)$ variance and PSQ has $O(\lambda_1^2)$ variance in this special case.

We extend this idea to the general case as the *block Householder quantizer* (BHQ). Specifically, we partition the rows into several groups. Within each group, only one row is large, and all the other rows are small. We apply the aforementioned Householder transformation separately for each group. The resultant scale matrix $\mathbf{S}^{(l)}$ is a block diagonal matrix, where each block is the product of a Householder matrix and a diagonal matrix. Again, BHQ takes $O(ND^{(l)})$ FP32 operations to implement. More details on the construction of BHQ are described in Appendix D.5.

## 4.3 Computational Overhead and Implementation

Here, we discuss the computational overhead and implementation details of our proposed quantizers. The actual time cost is highly platform-specific, and a complete hardware-algorithm co-design is out of the scope of this paper, which mostly focuses on the theoretical properties of gradient quantization. As a representative example, we investigate the quantization overhead for a $(N = 128, C = 64, H = W = 56)$ convolutional layer in INT8 on a single Intel CPU core, using a CPU version of TensorFlow [38] compiled with AVX support. In this case, the actual convolution takes 480ms. For quantization, we firstly need to find the dynamic range, which involves a per-tensor

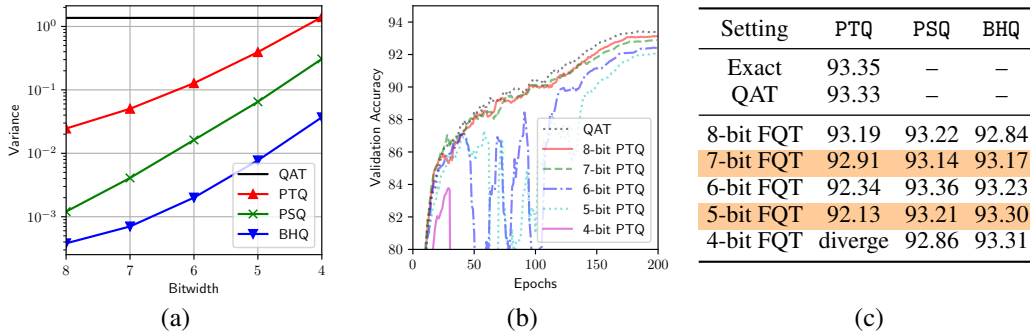

Figure 3: CIFAR10 convergence results. (a) Impact of gradient quantizer and bitwidth to gradient variance. (b) Convergence curve for PTQ. (c) Testing accuracy.

reduction of maximum and minimum elements for PTQ, and a per-sample reduction for PSQ and BHQ. Computing the range takes 11ms for PTQ and 24ms for PSQ and BHQ. The block Householder transformation can be implemented by two sparse-dense matrix multiplications. Suppose there are $G \leq N$ groups, each operation involves multiplying the gradient $\hat{\nabla}_{\mathbf{H}^{(l)}}$ by a $G \times N$ sparse matrix with $N$ non-zero elements. These matrix multiplications take $2ND^{(l)}$ FLOPs, or 21ms in total. Finally, we implement a custom C++ routine to find the optimal transformation and construct the sparse matrices for BHQ, which takes 3us. In general, the overhead for all the quantizers is small relative to the convolution.

Finally, dedicated hardware implementations may involve more subtleties. Instead of per-sample FP32 ranges, hardware may favor a per-tensor range with per-sample shift values ([0, 4] is enough according to our experience), where the accumulators are shifted before the final summation. We leave this as future work.

## 5 Empirical Evaluation

We demonstrate our theoretical results with an empirical evaluation on training ResNets [29] for image classification and transformers [39] for machine translation. Specifically, we use ResNet56-v2 [40] on the CIFAR10 [41] dataset, ResNet18/ResNet50 [29] for the ImageNet [42] dataset, and a transformer implemented in the Fairseq library [43] for the IWSLT14' En-DE machine translation dataset. The compared approaches are full-precision training (referred to as "exact"), QAT, and FQT. For QAT and FQT, we follow the settings on INT8 training [20]. Particularly, the activation and weights are quantized with PTQ to 8 bits. The gradient is quantized either with the baseline PTQ [20] or with our proposed PSQ and BHQ. In our setup, we keep the baseline PTQ mostly the same as [20], but we use batch normalization instead of range batch normalization. Furthermore, when computing the weight gradient $\hat{\nabla}_{\mathbf{\Theta}^{(l)}}$, we quantize the activation gradient $\hat{\nabla}_{\mathbf{H}^{(l)}}$ to 8-bit instead of leaving it in full precision. On the machine translation task, we only quantize all the linear layers for simplicity. Detailed experimental setup can be found in Appendix E.

### 5.1 Variance

Here, we first link the convergence properties of the algorithm with the gradient variance on the CIFAR10 dataset in Fig. 3. This illustrates that the quantization variance depends on the type of quantizer, as well as the bitwidth. Each fewer bit roughly increases the quantization variance by 4x, which aligns with our theoretical result. Moreover, BHQ achieves a similar variance as PTQ, with 3 fewer bits. When the quantization variance is relatively small (within 10% of the QAT variance), gradient quantization does not add much variance to the QAT gradient. This corresponds to PTQ with $\geq 7$ bits, PSQ with $\geq 5$ bits, and BHQ with $\geq 4$ bits. According to Fig. 3(b)(c), the validation accuracy degradation is small (within $0.4\%$) in these settings. On the other hand, the variance of PTQ is much larger than other quantizers, so its validation accuracy quickly decays and eventually diverges as the number of bits falls below 6. Therefore, gradient variance directly impacts the convergence.

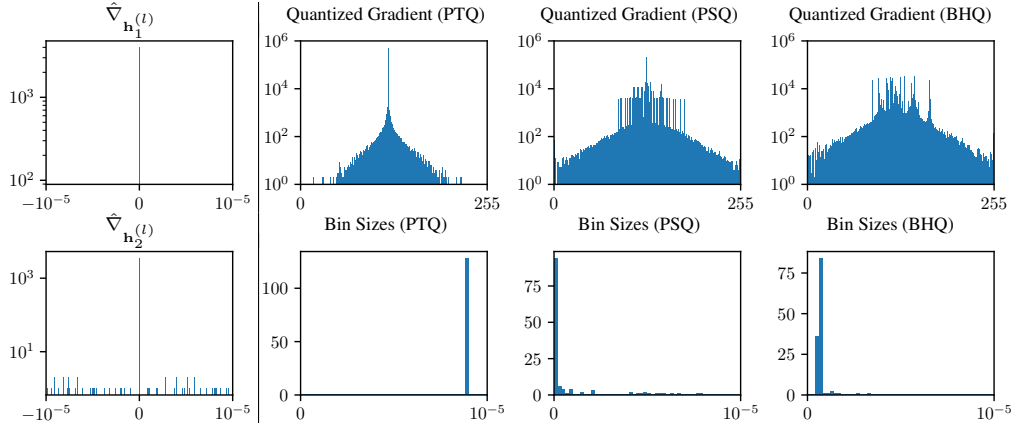

Figure 4: Histogram of gradients and quantization bin sizes. First row: PSQ and BHQ utilize more tail bins than PTQ. Second row: While all the quantization bins for PTQ are large, PSQ reduces the size of most bins and BHQ further eliminates all the large bins by distributing their values into smaller bins.

## 5.2 Variance Reduced Quantizers

Here, to illustrate the variance reduction effect of our gradient quantizers, we visualize the gradient at the `conv_3_5_2` layer on CIFAR-10 at the 100-th epoch in Fig. 4. The quantizer variance $\text{Var}\left[Q_b(\hat{\nabla}_{\mathbf{H}^{(l)}}) \mid \hat{\nabla}_{\mathbf{H}^{(l)}}\right]$ is $1.2 \times 10^{-3}$ for PTQ, $8.1 \times 10^{-5}$ for PSQ, and $1.4 \times 10^{-5}$ for BHQ. The gradients are quantized to $B = 255$ bins, and the histogram of the quantized gradients $\text{SR}\left(\mathbf{S}^{(l)}(\hat{\nabla}_{\mathbf{H}^{(l)}} - \mathbf{1}\mathbf{z}^{(l)})\right)$ are visualized in the first row of the right panel. In addition, we visualize the distribution of bin sizes in the second row, which is the actual numerical range that each quantization bin represents. Noticing that the bin size of PSQ is proportional to the dynamic range $R(\hat{\nabla}_{\mathbf{h}_i^{(L)}})$ of each row, we can observe that the dynamic range is close to zero for most correctly classified samples, but it is large for a few outliers. More concretely, we plot the histogram of the gradient of a correctly classified sample $\hat{\nabla}_{\mathbf{h}_1^{(l)}}$ and an outlier $\hat{\nabla}_{\mathbf{h}_2^{(l)}}$ on the left panel. This clearly shows that the gradient concentrates at zero for the correctly classified sample.

Since PTQ uses a single scale for the entire gradient, the quantized gradient is zero for most entries, showing a spike in the gradient histogram, and the utilization of other quantization bins are very low. The variance of PTQ is large, since it adopts a huge bin size across all entries. PSQ solves this problem by using separate scales for each sample, avoiding the unnecessarily large bin sizes for correctly classified data. As a result, its gradient histogram is much flatter, implying a better utilization of the bins on the tail. However, there are still some large quantization bins for the outliers. With the Householder transformation, BHQ splits the large gradients of outliers into other correctly classified samples. Therefore, the largest bin size of BHQ is much smaller than that of PSQ, at the expense of slightly increasing the bin size of correctly classified samples.

## 5.3 ImageNet Results

Here, we report both validation accuracy and training loss on ImageNet in Table 1 (convergence curves are in Appendix F). On ResNet18, our BHQ with a 5-bit gradient achieves $\leq 0.4\%$ validation accuracy degradation, comparable with the baseline BTQ with a 7-bit gradient. On the more challenging ResNet50, our PSQ and BHQ with an 8-bit gradient have indistinguishable results compared with QAT, while PTQ suffers from $\sim 1\%$ accuracy degradation. The accuracy degradation is still within $0.4\%$ for PSQ and BHQ with a 6-bit gradient. Our BHQ with 5-bit gradient performs as well as the baseline PTQ with an 8-bit gradient. Both PSQ and BHQ converge even with a 4-bit gradient, while PTQ diverges. Interestingly, the gain of BHQ and PSQ on ResNet50 is higher than that on ResNet18. We suspect it is due to the higher training accuracy on ResNet50, which makes the gradient sparser. We also compare our result with existing 8-bit training works in Table 2 in an end-to-end fashion. These results demonstrate that BHQ establishes a new state-of-the-art on this benchmark task.

Table 1: ResNet18/50 validation accuracy (training loss) on ImageNet.

| Setting | ResNet18 | | | ResNet50 | | |
|---|---|---|---|---|---|---|
| | PTQ | PSQ | BHQ | PTQ | PSQ | BHQ |
| Exact | 71.21 (2.20) | – | – | 77.09 (1.75) | – | – |
| QAT | 71.36 (2.25) | – | – | 77.35 (1.78) | – | – |
| 8-bit FQT | **71.24 (2.25)** | 70.92 (**2.25**) | 71.15 (**2.25**) | 76.40 (1.81) | **77.40 (1.77)** | 77.36 (**1.77**) |
| 7-bit FQT | 70.95 (2.26) | **71.00 (2.25)** | 70.85 (**2.25**) | 76.62 (1.80) | **77.36 (1.77)** | 76.96 (1.78) |
| 6-bit FQT | 70.73 (2.27) | 70.86 (**2.26**) | **71.01 (2.26)** | 76.06 (1.84) | 76.97 (1.79) | **77.25 (1.78)** |
| 5-bit FQT | 70.30 (2.30) | 70.57 (2.29) | **70.98 (2.27)** | 74.62 (1.93) | 76.30 (1.85) | **76.83 (1.81)** |
| 4-bit FQT | 68.70 (2.39) | 69.05 (2.39) | **69.48 (2.35)** | diverge | 73.78 (2.04) | **74.75 (1.96)** |

Table 2: 8-bit training results for ResNet50.

| Method | Val. acc. |
|---|---|
| FP8 [24] | 71.72 |
| HBFP8_16 [26] | 76.12 |
| HFP8 [25] | 76.46 |
| WAGEUBN [23] | 69.07 |
| Unified INT8 [22] | 76.34 |
| BHQ (ours) | **77.36** |

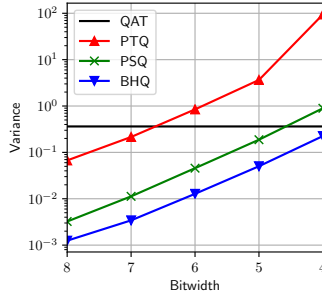

(a) Gradient variance

| Setting | PTQ | PSQ | BHQ |
|---|---|---|---|
| Exact | 34.55 | – | – |
| QAT | 34.47 | – | – |
| 8-bit | 34.33 | 34.39 | **34.51** |
| 5-bit | 0.02 | 33.17 | **33.70** |

(b) Validation BLEU score

Figure 5: Machine translation results on the IWSLT14' En-DE dataset. PSQ and BHQ achieve significantly lower gradient variance than PTQ, and converge even with a 5-bit gradient.

## 5.4 Machine Translation

Here, we report the validation BLEU score and the gradient variance on the machine translation task in Fig. 5. While the gradient variance increases exponentially as the bitwidth decreases, our BHQ and PSQ consistently achieves lower variance than the vanilla PTQ. Specifically, the gradient variance for 5-bit BHQ is roughly the same as that for 8-bit PTQ. In terms of validation BLUE score, while all the three gradient quantizers work well with 8-bit gradients, the vanilla PTQ diverges with 5-bit gradients. Meanwhile, BHQ still achieves a BLUE score within 1% degradation comparing with QAT. These observations are the same as those for image classification, indicating the general applicability of our approach.

## 6 Conclusions

We present a framework for FQT algorithms. Our framework assumes deterministic forward propagation and unbiased stochastic gradient quantizers. We formulate the FQT gradient as a stochastic estimator of the QAT gradient, and we derive its bias and variance, which impacts the convergence behavior of the training algorithm. To the best of our knowledge, this is the first result on how different gradient quantization schemes impact the gradient quality, without making strong assumptions such as single-layer network. Inspired by these theoretical results, we propose two novel gradient quantizers, PSQ and BHQ, and we demonstrate empirically that they have significantly lower variance than existing PTQ. Particularly, 5-bit BHQ performs as well as 8-bit PTQ for training ResNet50. There are many possible future directions based on this framework. Perhaps the most promising direction includes setting the gradient precision per layer adaptively, based on the variance; developing novel floating-point formats and vector quantizers; and developing theoretically inspired learning rate schedules.

## Broader Impact

Fully quantized training, including our work, can be potentially used to reduce the cost (and thus, for example, the carbon footprint) of training large deep neural networks. In recent years, huge models such as EfficientNet-B7 [16], BERT [18], GPT-2 [17] and GPT-3 [44] have achieve impressive results in many areas, particularly in natural language processing. However, these models are becoming prohibitively expensive to train. For example, the GPT-3 model takes 3,640 petaflops-days to train [44], while a V100 GPU only has ∼15 teraflops single precision throughput. Training is necessary when, for example, adapting to a new language. The prohibitive training time makes machine learning research and potential applications increasingly rely on these amounts of computational resources, and it is thus increasingly inaccessible and unfair. The low-bitwidth quantizers presented in this paper can potentially reduce the cost of training neural networks, making state-of-the-art machine learning more democratized. Fully quantized training may also be applied for training on edge devices. Due to the high energy cost, training is not yet widely done on edge devices. Using the energy-efficient low-bitwidth hardware, the techniques proposed in this paper can potentially help move training towards edge. Training on edge enables new applications, such as locally-trained personalized models. Locally trained models improve privacy, as they do not need to upload user information to the cloud.

## Acknowlegements

We thank Liam Hodgkinson and Amir Gholami for proofreading and helpful discussions. This work was supported by a gracious fund from Intel corporation, Berkeley Deep Drive (BDD), and Berkeley AI Research (BAIR) sponsors. In addition to NSF CISE Expeditions Award CCF-1730628, this research is supported by gifts from Amazon Web Services, Ant Group, CapitalOne, Ericsson, Facebook, Futurewei, Google, Intel, Microsoft, Nvidia, Scotiabank, Splunk and VMware. We would like to thank the Intel VLAB team for providing us with access to their computing cluster. We also gratefully acknowledge the support of NVIDIA Corporation for their donation of two Titan Xp GPU used for this research. We would also like to acknowledge the UC Berkeley CLTC, ARO, IARPA, NSF, and ONR for providing partial support of this work. Our conclusions do not necessarily reflect the position or the policy of our sponsors, and no official endorsement should be inferred.

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
