[Supplementary Material]

# A Table of Notations

Table 3: Table of Notations.

| Notation | Description |
|---|---|
| $\mathbf{X}$ | A batch of inputs (each row is a sample) |
| $\mathbf{Y}$ | A batch of labels (each row is a sample) |
| $\mathcal{B}$ | A batch $\mathcal{B} = (\mathbf{X}, \mathbf{Y})$ |
| $N, C, L$ | Batch size, number of classes, and number of layers |
| $Q_f(\cdot), Q_\theta(\cdot), Q_b(\cdot)$ | activation / parameter / gradient quantizer |
| $\mathbf{F}(\cdot; \boldsymbol{\Theta})$ | DNN with parameter $\boldsymbol{\Theta}$ |
| $\mathbf{F}^{(l)}(\cdot; \boldsymbol{\Theta}^{(l)})$ | $l$-th layer with parameter $\boldsymbol{\Theta}^{(l)}$ |
| $\mathbf{H}^{(l)}$ | Activation matrix at layer $l$, whose size is $N \times D^{(l)}$ |
| $\tilde{\mathbf{H}}^{(l)}, \tilde{\boldsymbol{\Theta}}^{(l)}$ | Quantized activation / parameter |
| $\ell(\mathbf{H}^{(L)}, \mathcal{Y})$ | loss function of prediction $\mathbf{H}^{(L)}$ and label $\mathcal{Y}$. |
| $\nabla_{\boldsymbol{\Theta}} \ell$ | Gradient of $\ell$ w.r.t. $\boldsymbol{\Theta}$ |
| $\mathbf{J}^{(l)}$ | Jacobian matrix $\frac{\partial \text{vec}(\mathbf{H}^{(l)})}{\partial \text{vec}(\tilde{\mathbf{H}}^{(l-1)})}$ |
| $\mathbf{K}^{(l)}$ | Jacobian matrix $\frac{\partial \text{vec}(\mathbf{H}^{(l)})}{\partial \text{vec}(\tilde{\boldsymbol{\Theta}}^{(l)})}$ |
| $\nabla_{\mathbf{H}^{(l)}}, \nabla_{\boldsymbol{\Theta}^{(l)}}, \nabla_{\boldsymbol{\Theta}}$ | QAT gradient for activation / parameter |
| $\hat{\nabla}_{\mathbf{H}^{(l)}}, \hat{\nabla}_{\boldsymbol{\Theta}^{(l)}}, \hat{\nabla}_{\boldsymbol{\Theta}}$ | FQT gradient for activation / parameter |
| $\nabla_{\mathbf{h}_i^{(l)}}, \hat{\nabla}_{\mathbf{h}_i^{(l)}}$ | $i$-th row of QAT / FQT activation gradient at $l$-th layer |
| $\mathbb{E}[X \mid Y]$ | Conditional expectation of $X$ given $Y$ |
| $\text{Var}[X \mid Y]$ | Conditional variance of $X$ given $Y$ |
| $R(\mathbf{X})$ | Dynamic range of $\mathbf{X}$, i.e., $\max \mathbf{X} - \min \mathbf{X}$ |
| $b, B$ | Number of quantization bits / bins |

# B Preliminary Knowledge

**Proposition 1.** *(Law of total variance) If* $\mathbf{X}$ *and* $\mathbf{Y}$ *are random matrices on the same probability space, and all elements of* $\text{Var}[\mathbf{Y}]$ *is finite, then*

$$\text{Var}[\mathbf{Y}] = \mathbb{E}[\text{Var}[\mathbf{Y} \mid \mathbf{X}]] + \text{Var}[\mathbb{E}[\mathbf{Y} \mid \mathbf{X}]].$$

*Proof.* By the definition of variance,

$$\text{Var}[\mathbf{Y}] = \sum_{ij} \mathbb{E}[Y_{ij}^2] - \mathbb{E}[Y_{ij}]^2.$$

By law of total expectation,

$$
\begin{aligned}
\mathbb{E}[Y_{ij}^2] - \mathbb{E}[Y_{ij}]^2 &= \mathbb{E}[\mathbb{E}[Y_{ij}^2 \mid \mathbf{X}]] - \mathbb{E}[\mathbb{E}[Y_{ij} \mid \mathbf{X}]]^2 \\
&= \mathbb{E}[\text{Var}[Y_{ij} \mid \mathbf{X}] + \mathbb{E}[Y_{ij} \mid \mathbf{X}]^2] - \mathbb{E}[\mathbb{E}[Y_{ij} \mid \mathbf{X}]]^2 \\
&= \mathbb{E}[\text{Var}[Y_{ij} \mid \mathbf{X}]] + \mathbb{E}[\mathbb{E}[Y_{ij} \mid \mathbf{X}]^2] - \mathbb{E}[\mathbb{E}[Y_{ij} \mid \mathbf{X}]]^2 \\
&= \mathbb{E}[\text{Var}[Y_{ij} \mid \mathbf{X}]] + \text{Var}[\mathbb{E}[Y_{ij} \mid \mathbf{X}]].
\end{aligned}
$$

Putting it together, we have

$$\text{Var}[\mathbf{Y}] = \sum_{ij} \mathbb{E}[\text{Var}[Y_{ij} \mid \mathbf{X}]] + \text{Var}[\mathbb{E}[Y_{ij} \mid \mathbf{X}]] = \mathbb{E}[\text{Var}[\mathbf{Y} \mid \mathbf{X}]] + \text{Var}[\mathbb{E}[\mathbf{Y} \mid \mathbf{X}]].$$

$\square$

**Proposition 2.** *For a random matrix* $\mathbf{X}$ *and a constant matrix* $\mathbf{W}$,

$$\text{Var}[\mathbf{X}\mathbf{W}] \le \text{Var}[\mathbf{X}] \|\mathbf{W}\|_2^2.$$

*Proof.* Firstly, for any matrices $\mathbf{A}$ and $\mathbf{B}$, by the definition of Frobenius and operator norm, we have

$$\|\mathbf{AB}\|_F^2 = \sum_i \|\mathbf{a}_i\mathbf{B}\|_2^2 \le \sum_i \|\mathbf{a}_i\|_2^2 \|\mathbf{B}\|_2^2 = \|\mathbf{A}\|_F^2 \|\mathbf{B}\|_2^2.$$

Let $\boldsymbol{\mu} = \mathbb{E}[\mathbf{X}]$, and utilize this inequality, we have

$$\mathrm{Var}[\mathbf{XW}] = \mathbb{E}\|\mathrm{vec}(\mathbf{XW}) - \mathbb{E}[\mathrm{vec}(\mathbf{XW})]\|_2^2 = \mathbb{E}\|\mathbf{XW} - \mathbb{E}[\mathbf{XW}]\|_F^2$$
$$= \mathbb{E}\|(\mathbf{X} - \boldsymbol{\mu})\mathbf{W}\|_F^2 \le \mathbb{E}\left[\|(\mathbf{X} - \boldsymbol{\mu})\|_F^2 \|\mathbf{W}\|_2^2\right] = \mathrm{Var}[\mathbf{X}] \|\mathbf{W}\|_2^2.$$

$\square$

**Proposition 3.** *For constant matrices $\mathbf{A}$, $\mathbf{B}$ and a random matrix $\boldsymbol{\epsilon}$, if for all entries $i, j$, $\mathrm{Var}[\epsilon_{ij}] \le \sigma^2$, then*

$$\mathrm{Var}[\mathbf{A}\boldsymbol{\epsilon}\mathbf{B}] \le \sigma^2 \|\mathbf{A}\|_F^2 \|\mathbf{B}\|_F^2.$$

*Proof.*

$$\mathrm{Var}[\mathbf{A}\boldsymbol{\epsilon}\mathbf{B}] = \sum_{ij} \mathrm{Var}[\mathbf{a}_i\boldsymbol{\epsilon}\mathbf{B}_{:,j}] = \sum_{ij} \mathrm{Var}\left[\sum_{kl} A_{ik}\epsilon_{kl}B_{lj}\right] = \sum_{ijkl} A_{ik}^2 \mathrm{Var}[\epsilon_{kl}] B_{lj}^2$$
$$\le \sigma^2 \sum_{ijkl} A_{ik}^2 B_{lj}^2 = \sigma^2 \|\mathbf{A}\|_F^2 \|\mathbf{B}\|_F^2.$$

$\square$

## C  Proofs

In this section, we give the proofs on the gradient bias and variance used in the main text.

### C.1  Proof of Theorem 1

*Proof.* We prove by induction. Firstly,

$$\hat{\nabla}_{\mathbf{H}^{(L)}} = \nabla_{\mathbf{H}^{(L)}} = \partial\ell/\partial\mathbf{H}^{(L)},$$

so $\mathbb{E}\left[\hat{\nabla}_{\mathbf{H}^{(l)}} \mid \mathcal{B}\right] = \nabla_{\mathbf{H}^{(l)}}$ holds for $l = L$. Assume that $\mathbb{E}\left[\hat{\nabla}_{\mathbf{H}^{(l)}} \mid \mathcal{B}\right] = \nabla_{\mathbf{H}^{(l)}}$ holds for $l$, then we have

$$\mathrm{vec}\left(\mathbb{E}\left[\hat{\nabla}_{\mathbf{H}^{(l-1)}} \mid \mathcal{B}\right]\right) = \mathbb{E}\left[\mathrm{vec}(\hat{\nabla}_{\mathbf{H}^{(l-1)}}) \mid \mathcal{B}\right],$$

because $\mathrm{vec}(\cdot)$ does not affect the expectation. According to the definition Eq. (5), we have

$$\mathbb{E}\left[\mathrm{vec}(\hat{\nabla}_{\mathbf{H}^{(l-1)}}) \mid \mathcal{B}\right] = \mathbb{E}\left[\mathrm{vec}(Q_b(\hat{\nabla}_{\mathbf{H}^{(l)}}))\mathbf{J}^{(l)} \mid \mathcal{B}\right].$$

Since $\mathbf{J}^{(l)}$ is deterministic given $\mathcal{B}$, we have

$$\mathbb{E}\left[\mathrm{vec}(Q_b(\hat{\nabla}_{\mathbf{H}^{(l)}}))\mathbf{J}^{(l)} \mid \mathcal{B}\right] = \mathrm{vec}\left(\mathbb{E}\left[Q_b(\hat{\nabla}_{\mathbf{H}^{(l)}}) \mid \mathcal{B}\right]\right)\mathbf{J}^{(l)} = \mathrm{vec}\left(\mathbb{E}\left[\hat{\nabla}_{\mathbf{H}^{(l)}} \mid \mathcal{B}\right]\right)\mathbf{J}^{(l)}.$$

By induction assumption and Eq. (4),

$$\mathrm{vec}\left(\mathbb{E}\left[\hat{\nabla}_{\mathbf{H}^{(l)}} \mid \mathcal{B}\right]\right)\mathbf{J}^{(l)} = \mathrm{vec}(\nabla_{\mathbf{H}^{(l)}})\mathbf{J}^{(l)} = \mathrm{vec}(\nabla_{\mathbf{H}^{(l-1)}}).$$

So $\mathbb{E}\left[\hat{\nabla}_{\mathbf{H}^{(l-1)}} \mid \mathcal{B}\right] = \nabla_{\mathbf{H}^{(l-1)}}$. Similarly,

$$\mathrm{vec}\left(\mathbb{E}\left[\hat{\nabla}_{\boldsymbol{\Theta}^{(l)}} \mid \mathcal{B}\right]\right) = \mathbb{E}\left[\mathrm{vec}(Q_b(\hat{\nabla}_{\mathbf{H}^{(l)}}))\mathbf{K}^{(l)} \mid \mathcal{B}\right] = \mathrm{vec}(\nabla_{\mathbf{H}^{(l)}})\mathbf{K}^{(l)} = \mathrm{vec}(\nabla_{\boldsymbol{\Theta}^{(l)}}).$$

Therefore, $\mathbb{E}\left[\hat{\nabla}_{\boldsymbol{\Theta}^{(l)}} \mid \mathcal{B}\right] = \nabla_{\boldsymbol{\Theta}^{(l)}}$. Taking $l$ from $L$ to 1, we prove

$$\forall l \in [L], \mathbb{E}\left[\hat{\nabla}_{\mathbf{H}^{(l)}} \mid \mathcal{B}\right] = \nabla_{\mathbf{H}^{(l)}}; \quad \forall l \in [L]_+, \mathbb{E}\left[\hat{\nabla}_{\boldsymbol{\Theta}^{(l)}} \mid \mathcal{B}\right] = \nabla_{\boldsymbol{\Theta}^{(l)}},$$

so $\mathbb{E}\left[\hat{\nabla}_{\boldsymbol{\Theta}} \mid \mathcal{B}\right] = \nabla_{\boldsymbol{\Theta}}$.

$\square$

## C.2 Proof of Theorem 2

*Proof.* By Proposition 1 and Theorem 1, we have

$$\text{Var}\left[\hat{\nabla}_{\boldsymbol{\Theta}}\right] = \mathbb{E}\left[\text{Var}\left[\hat{\nabla}_{\boldsymbol{\Theta}} \,\middle|\, \mathcal{B}\right]\right] + \text{Var}\left[\mathbb{E}\left[\hat{\nabla}_{\boldsymbol{\Theta}} \,\middle|\, \mathcal{B}\right]\right] = \mathbb{E}\left[\text{Var}\left[\hat{\nabla}_{\boldsymbol{\Theta}} \,\middle|\, \mathcal{B}\right]\right] + \text{Var}\left[\nabla_{\boldsymbol{\Theta}}\right].$$

By definition of $\text{Var}\left[\cdot\right]$, we have $\text{Var}\left[\hat{\nabla}_{\boldsymbol{\Theta}} \,\middle|\, \mathcal{B}\right] = \sum_{l=1}^{L}\text{Var}\left[\text{vec}(\hat{\nabla}_{\boldsymbol{\Theta}^{(l)}}) \,\middle|\, \mathcal{B}\right]$. Apply Proposition 1 and Eq. (5), we have

$$\mathbb{E}\left[\text{Var}\left[\text{vec}(\hat{\nabla}_{\boldsymbol{\Theta}^{(l)}}) \,\middle|\, \mathcal{B}\right]\right]$$

$$=\mathbb{E}\left[\text{Var}\left[\text{vec}(Q_b(\hat{\nabla}_{\mathbf{H}^{(l)}}))\mathbf{K}^{(l)} \,\middle|\, \mathcal{B}\right]\right]$$

$$=\mathbb{E}\left[\text{Var}\left[\text{vec}(Q_b(\hat{\nabla}_{\mathbf{H}^{(l)}}))\mathbf{K}^{(l)} \,\middle|\, \hat{\nabla}_{\mathbf{H}^{(l)}}\right]\right] + \mathbb{E}\left[\text{Var}\left[\mathbb{E}\left[\text{vec}(Q_b(\hat{\nabla}_{\mathbf{H}^{(l)}}))\mathbf{K}^{(l)} \,\middle|\, \hat{\nabla}_{\mathbf{H}^{(l)}}\right] \,\middle|\, \mathcal{B}\right]\right]$$

$$=\mathbb{E}\left[\text{Var}\left[\text{vec}(Q_b(\hat{\nabla}_{\mathbf{H}^{(l)}}))\mathbf{K}^{(l)} \,\middle|\, \hat{\nabla}_{\mathbf{H}^{(l)}}\right]\right] + \mathbb{E}\left[\text{Var}\left[\text{vec}(\hat{\nabla}_{\mathbf{H}^{(l)}})\mathbf{K}^{(l)} \,\middle|\, \mathcal{B}\right]\right],$$

where

$$\mathbb{E}\left[\text{Var}\left[\text{vec}(\hat{\nabla}_{\mathbf{H}^{(l)}})\mathbf{K}^{(l)} \,\middle|\, \mathcal{B}\right]\right]$$

$$=\mathbb{E}\left[\text{Var}\left[\text{vec}(Q_b(\hat{\nabla}_{\mathbf{H}^{(l+1)}}))\mathbf{J}^{(l+1)}\mathbf{K}^{(l)} \,\middle|\, \hat{\nabla}_{\mathbf{H}^{(l+1)}}\right]\right] + \mathbb{E}\left[\text{Var}\left[\mathbb{E}\left[\text{vec}(Q_b(\hat{\nabla}_{\mathbf{H}^{(l+1)}}))\mathbf{J}^{(l+1)}\mathbf{K}^{(l)} \,\middle|\, \hat{\nabla}_{\mathbf{H}^{(l+1)}}\right] \,\middle|\, \mathcal{B}\right]\right]$$

$$=\mathbb{E}\left[\text{Var}\left[\text{vec}(Q_b(\hat{\nabla}_{\mathbf{H}^{(l+1)}}))\mathbf{J}^{(l+1)}\mathbf{K}^{(l)} \,\middle|\, \hat{\nabla}_{\mathbf{H}^{(l+1)}}\right]\right] + \mathbb{E}\left[\text{Var}\left[\text{vec}(\hat{\nabla}_{\mathbf{H}^{(l+1)}})\mathbf{J}^{(l+1)}\mathbf{K}^{(l)} \,\middle|\, \mathcal{B}\right]\right].$$

Repeat this procedure, we can finally get

$$\mathbb{E}\left[\text{Var}\left[\text{vec}(\hat{\nabla}_{\boldsymbol{\Theta}^{(l)}}) \,\middle|\, \mathcal{B}\right]\right] = \sum_{k=l}^{L}\mathbb{E}\left[\text{Var}\left[\text{vec}(Q_b(\hat{\nabla}_{\mathbf{H}^{(k)}}))\boldsymbol{\gamma}^{(l,k)} \,\middle|\, \hat{\nabla}_{\mathbf{H}^{(k)}}\right]\right].$$

Putting it together, we have

$$\text{Var}\left[\hat{\nabla}_{\boldsymbol{\Theta}}\right] =\mathbb{E}\left[\text{Var}\left[\hat{\nabla}_{\boldsymbol{\Theta}} \,\middle|\, \mathcal{B}\right]\right] + \text{Var}\left[\nabla_{\boldsymbol{\Theta}}\right] = \text{Var}\left[\nabla_{\boldsymbol{\Theta}}\right] + \sum_{l=1}^{L}\mathbb{E}\left[\text{Var}\left[\text{vec}(\hat{\nabla}_{\boldsymbol{\Theta}^{(l)}}) \,\middle|\, \mathcal{B}\right]\right]$$

$$=\text{Var}\left[\nabla_{\boldsymbol{\Theta}}\right] + \sum_{l=1}^{L}\sum_{k=l}^{L}\mathbb{E}\left[\text{Var}\left[\text{vec}(Q_b(\hat{\nabla}_{\mathbf{H}^{(k)}}))\boldsymbol{\gamma}^{(l,k)} \,\middle|\, \hat{\nabla}_{\mathbf{H}^{(k)}}\right]\right]$$

$$=\text{Var}\left[\nabla_{\boldsymbol{\Theta}}\right] + \sum_{k=1}^{L}\sum_{l=1}^{k}\mathbb{E}\left[\text{Var}\left[\text{vec}(Q_b(\hat{\nabla}_{\mathbf{H}^{(k)}}))\boldsymbol{\gamma}^{(l,k)} \,\middle|\, \hat{\nabla}_{\mathbf{H}^{(k)}}\right]\right]$$

$$=\text{Var}\left[\nabla_{\boldsymbol{\Theta}}\right] + \sum_{l=1}^{L}\mathbb{E}\left[\sum_{k=1}^{l}\text{Var}\left[\text{vec}(Q_b(\hat{\nabla}_{\mathbf{H}^{(l)}}))\boldsymbol{\gamma}^{(k,l)} \,\middle|\, \hat{\nabla}_{\mathbf{H}^{(l)}}\right]\right], \qquad (7)$$

where in the second last line we swap the order of inner and outer summations, and in the last line we swap the symbols $k$ and $l$, and utilize the linearity of expectation.

Utilizing Proposition 2, we have

$$\text{Var}\left[\text{vec}(Q_b(\hat{\nabla}_{\mathbf{H}^{(l)}}))\boldsymbol{\gamma}^{(k,l)} \,\middle|\, \hat{\nabla}_{\mathbf{H}^{(l)}}\right] \leq \text{Var}\left[\text{vec}(Q_b(\hat{\nabla}_{\mathbf{H}^{(l)}})) \,\middle|\, \hat{\nabla}_{\mathbf{H}^{(l)}}\right]\left\|\boldsymbol{\gamma}^{(k,l)}\right\|_2^2.$$

Putting it together

$$\text{Var}\left[\hat{\nabla}_{\boldsymbol{\Theta}}\right] \leq \text{Var}\left[\nabla_{\boldsymbol{\Theta}}\right] + \sum_{l=1}^{L}\mathbb{E}\left[\sum_{k=1}^{l}\text{Var}\left[\text{vec}(Q_b(\hat{\nabla}_{\mathbf{H}^{(l)}})) \,\middle|\, \hat{\nabla}_{\mathbf{H}^{(l)}}\right]\left\|\boldsymbol{\gamma}^{(k,l)}\right\|_2^2\right]$$

$$= \text{Var}\left[\nabla_{\boldsymbol{\Theta}}\right] + \sum_{l=1}^{L}\mathbb{E}\left[\text{Var}\left[Q_b(\hat{\nabla}_{\mathbf{H}^{(l)}}) \,\middle|\, \hat{\nabla}_{\mathbf{H}^{(l)}}\right]\sum_{k=1}^{l}\left\|\boldsymbol{\gamma}^{(k,l)}\right\|_2^2\right].$$

$\square$

# D  Variance of Specific Quantizers

**Proposition 4.** *(Variance of stochastic rounding) For any* $\mathbf{X} \in \mathbb{R}^{N \times M}$, $\mathrm{Var}\left[\mathrm{SR}(\mathbf{X})\right] \leq \frac{NM}{4}$.

*Proof.* For any real number $X$, let $p := X - \lfloor X \rfloor \in [0, 1)$, then

$$\mathrm{Var}\left[\mathrm{SR}(X)\right] = \mathbb{E}[\mathrm{SR}(X) - X]^2 = p(\lceil X \rceil - X)^2 + (1 - p)(\lfloor X \rfloor - X)^2$$

$$=p(1 - p)^2 + p^2(1 - p) = p(1 - p)(1 - p + p) = p(1 - p) \leq \frac{1}{4}.$$

Therefore, according to Definition 1,

$$\mathrm{Var}\left[\mathrm{SR}(\mathbf{X})\right] = \sum_{ij} \mathrm{Var}\left[\mathrm{SR}(X_{ij})\right] = \frac{NM}{4}.$$

$\square$

For simplicity, all the expectation and variance are conditioned on $\hat{\nabla}_{\mathbf{H}^{(l)}}$ in the rest of this section.

## D.1  Per-tensor Quantizer

$$\mathrm{Var}\left[Q_b(\hat{\nabla}_{\mathbf{H}^{(l)}})\right] = \mathrm{Var}\left[\mathrm{SR}\left(S^{(l)}(\hat{\nabla}_{\mathbf{H}^{(l)}} - Z^{(l)})\right) / S^{(l)} + Z^{(l)}\right]$$

$$=\frac{1}{(S^{(l)})^2}\mathrm{Var}\left[\mathrm{SR}\left(S^{(l)}(\hat{\nabla}_{\mathbf{H}^{(l)}} - Z^{(l)})\right)\right] \leq \frac{ND^{(l)}}{4(S^{(l)})^2} = \frac{ND^{(l)}}{4B^2}R(\hat{\nabla}_{\mathbf{H}^{(l)}})^2.$$

## D.2  Matrix Quantizer

For the matrix quantizer defined in Eq. (11), we have

$$\mathrm{Var}\left[Q_b(\hat{\nabla}_{\mathbf{H}^{(l)}})\right] = \mathrm{Var}\left[(\mathbf{S}^{(l)})^{-1}\mathrm{SR}\left(\mathbf{S}^{(l)}(\hat{\nabla}_{\mathbf{H}^{(l)}} - \mathbf{1}\mathbf{z}^{(l)})\right) + \mathbf{1}\mathbf{z}^{(l)}\right] = \mathrm{Var}\left[(\mathbf{S}^{(l)})^{-1}\mathrm{SR}\left(\mathbf{S}^{(l)}(\hat{\nabla}_{\mathbf{H}^{(l)}} - \mathbf{1}\mathbf{z}^{(l)})\right)\right].$$

Utilizing Proposition 3 with $\mathbf{A} = (\mathbf{S}^{(l)})^{-1}$, $\boldsymbol{\epsilon} = \mathrm{SR}\left(\mathbf{S}^{(l)}(\hat{\nabla}_{\mathbf{H}^{(l)}} - \mathbf{1}\mathbf{z}^{(l)})\right)$, and $\mathbf{B} = \mathbf{I}$,

$$\mathrm{Var}\left[Q_b(\hat{\nabla}_{\mathbf{H}^{(l)}})\right] \leq \frac{1}{4}\left\|(\mathbf{S}^{(l)})^{-1}\right\|_F^2 \|\mathbf{I}\|_F^2 = \frac{D^{(l)}}{4}\left\|(\mathbf{S}^{(l)})^{-1}\right\|_F^2. \tag{13}$$

Minimizing Eq. (13) w.r.t. $\mathbf{S}^{(l)}$ yields optimization problem (12) as follows

$$\min_{\mathbf{S}^{(l)}}\left\|(\mathbf{S}^{(l)})^{-1}\right\|_F^2, \text{ s.t. } R(\mathbf{S}^{(l)}\hat{\nabla}_{\mathbf{H}^{(l)}}) \leq B,$$

## D.3  Per-sample Quantizer

When $\mathbf{S} = \mathrm{diag}(s_1, \dots, s_N)$, we can rewrite optimization problem (12) as

$$\min_{s_1,\dots,s_N} \sum_{i=1}^N s_i^{-2}, \text{ s.t. } s_i R(\hat{\nabla}_{\mathbf{h}_i^{(l)}}) \leq B, \forall i \in [N]_+. \tag{14}$$

Since the objective is monotonic w.r.t. $s_i$, problem (14) can be minimized when all the inequality constraints takes equality, i.e., $s_i R(\hat{\nabla}_{\mathbf{h}_i^{(l)}}) = B$. Therefore, $s_i = B/R(\hat{\nabla}_{\mathbf{h}_i^{(l)}})$. Plug this back to Eq. (13), we have

$$\mathrm{Var}\left[Q_b(\hat{\nabla}_{\mathbf{H}^{(l)}})\right] \leq \frac{D^{(l)}}{4}\left\|(\mathbf{S}^{(l)})^{-1}\right\|_F^2 = \frac{D^{(l)}}{4}\sum_{i=1}^N\left(B/R(\hat{\nabla}_{\mathbf{h}_i^{(l)}})\right)^{-2} = \frac{D^{(l)}}{4B^2}\sum_{i=1}^N R(\hat{\nabla}_{\mathbf{h}_i^{(l)}})^2.$$

## D.4 Householder Quantizer

Let $\lambda_1 = R(\hat{\nabla}_{\mathbf{h}_1^{(L)}})$, $\lambda_2 = 2\max_{i\neq 1}\left\|\hat{\nabla}_{\mathbf{h}_i^{(L)}}\right\|_\infty$, and assume $\lambda_2/\lambda_1 \approx 0$. Without loss of generality, we can write

$$\hat{\nabla}_{\mathbf{H}^{(l)}} = \begin{bmatrix} \hat{\nabla}_{\mathbf{h}_1^{(l)}} \\ \hat{\nabla}_{\mathbf{H}_{>1}^{(l)}} \end{bmatrix} = \begin{bmatrix} \hat{\nabla}_{\mathbf{h}_1^{(l)}} \\ \mathbf{0} \end{bmatrix} + \begin{bmatrix} \mathbf{0} \\ \hat{\nabla}_{\mathbf{H}_{>1}^{(l)}} \end{bmatrix} = \lambda_1 \mathbf{e}_1 \mathbf{u}_1 + \frac{1}{2}\lambda_2 \mathbf{U}_2,$$

such that $R(\mathbf{u}_1) \leq 1$, and $\max_{i\neq 1}\left\|\hat{\nabla}_{\mathbf{h}_i^{(L)}}\right\|_\infty \leq 1$, and $\mathbf{e}_1$ is a column coordinate vector. Furthermore, we construct $\mathbf{S}^{(l)} = \mathbf{Q}\mathrm{diag}(s_1, s_2, \ldots, s_2)$, where $\mathbf{Q} = \mathbf{I} - 2\mathbf{n}\mathbf{n}^\top/\|\mathbf{n}\|_2^2$ is a Householder reflection with the normal vector $\mathbf{n} = \mathbf{1}/\sqrt{N} - \mathbf{e}_1$.

We have

$$\mathbf{S}^{(l)}\hat{\nabla}_{\mathbf{H}^{(l)}} = \mathbf{Q}\mathrm{diag}(s_1, s_2, \ldots, s_2)\left(\lambda_1 \mathbf{e}_1 \mathbf{u}_1 + \frac{1}{2}\lambda_2 \mathbf{U}_2\right) = \mathbf{Q}\left(\lambda_1 s_1 \mathbf{e}_1 \mathbf{u}_1 + \frac{1}{2}\lambda_2 s_2 \mathbf{U}_2\right)$$

$$= \lambda_1 s_1 N^{-1/2}\mathbf{1}\mathbf{u}_1 + \frac{1}{2}\lambda_2 s_2 \mathbf{Q}\mathbf{U}_2.$$

Then, utilizing $R(\mathbf{u}_1) \leq 1$,

$$R(\lambda_1 s_1 N^{-1/2}\mathbf{1}\mathbf{u}_1) = \lambda_1 s_1 N^{-1/2}R(\mathbf{1}\mathbf{u}_1) = \lambda_1 s_1 N^{-1/2}(\max_j \mathbf{u}_{1j} - \min_j \mathbf{u}_{1j}) \leq \lambda_1 s_1 N^{-1/2}.$$

On the other hand,

$$R(\frac{1}{2}\lambda_2 s_2 \mathbf{Q}\mathbf{U}_2) = \frac{1}{2}\lambda_2 s_2 R(\mathbf{Q}\mathbf{U}_2) \leq \lambda_2 s_2 \|\mathbf{Q}\mathbf{U}_2\|_\infty = \lambda_2 s_2 \max_j \|\mathbf{Q}\mathbf{U}_{2,:j}\|_\infty,$$

and

$$\|\mathbf{Q}\mathbf{U}_{2,:j}\|_\infty \leq \|\mathbf{Q}\mathbf{U}_{2,:j}\|_2 = \|\mathbf{U}_{2,:j}\|_2 \leq \sqrt{N}\|\mathbf{U}_{2,:j}\|_\infty \leq \sqrt{N}.$$

Putting it together, we have

$$R(\mathbf{S}^{(l)}\hat{\nabla}_{\mathbf{H}^{(l)}}) \leq R(\lambda_1 s_1 N^{-1/2}\mathbf{1}\mathbf{u}_1) + R(\frac{1}{2}\lambda_2 s_2 \mathbf{Q}\mathbf{U}_2) \leq \lambda_1 s_1 N^{-1/2} + \lambda_2 s_2 N^{1/2}.$$

Therefore, problem (12) can be rewritten as

$$\min_{s_1, s_2} s_1^{-2} + (N-1)s_2^{-2}, \quad \text{s.t. } \lambda_1 s_1 N^{-1/2} + \lambda_2 s_2 N^{1/2} = B.$$

We minimize an upper bound instead

$$\min_{s_1, s_2} s_1^{-2} + Ns_2^{-2}, \quad \text{s.t. } \lambda_1 s_1 N^{-1/2} + \lambda_2 s_2 N^{1/2} = B.$$

Introducing the multiplier $\tau$, and define the Lagrangian

$$f(s_1, s_2, \tau) = s_1^{-2} + Ns_2^{-2} + \tau\left(\lambda_1 s_1 N^{-1/2} + \lambda_2 s_2 N^{1/2} - B\right).$$

Letting $\partial f/\partial s_1 = \partial f/\partial s_2 = 0$, we have

$$-2s_1^{-3} + \tau\lambda_1 N^{-1/2} = 0 \Rightarrow s_1 \propto \lambda_1^{-1/3}N^{1/6}$$

$$-2Ns_2^{-3} + \tau\lambda_2 N^{1/2} = 0 \Rightarrow s_2 \propto \lambda_2^{-1/3}N^{1/6},$$

utilizing the equality constraint $\lambda_1 s_1 N^{-1/2} + \lambda_2 s_2 N^{1/2} = B$, we have

$$s_1 = B\frac{\lambda_1^{-1/3}N^{1/6}}{\lambda_1^{2/3}N^{-1/3} + \lambda_2^{2/3}N^{2/3}}, \quad s_2 = B\frac{\lambda_2^{-1/3}N^{1/6}}{\lambda_1^{2/3}N^{-1/3} + \lambda_2^{2/3}N^{2/3}}.$$

Therefore, we have

$$\left\|(\mathbf{S}^{(l)})^{-1}\right\|_F^2 = s_1^{-2} + (N-1)s_2^{-2} < s_1^{-2} + Ns_2^{-2} = \frac{1}{B^2}\left(\lambda_1^{2/3}N^{-1/3} + \lambda_2^{2/3}N^{2/3}\right)^3,$$

plugging it to Eq. (13), we have

$$\mathrm{Var}\left[Q_b(\hat{\nabla}_{\mathbf{H}^{(l)}})\right] \leq \frac{D^{(l)}}{4B^2}\left(\lambda_1^{2/3}N^{-1/3} + \lambda_2^{2/3}N^{2/3}\right)^3 \approx \frac{D^{(l)}}{4B^2}\lambda_1^2 N^{-1} = O(\lambda_1^2/N).$$

## D.5 Details of Block Householder Quantizer

We construct the block Householder quantizer as follows.

1. Sort the magnitude $M_i := \left\| \hat{\nabla}_{\mathbf{h}_i^{(l)}} \right\|_\infty$ of each row in descending order.

2. Loop over the number of groups $G$. Assume that $\{M_i\}$ is already sorted, we consider the first $G$ rows as "large" and all the other $N - G$ rows as "small". The $i$-th group contains the $i$-th largest row and a number of small rows. Furthermore, we heuristically set the size of the $i$-th group to $(N - G)\frac{M_i}{\sum_{i=1}^{G} M_i}$, i.e., proportional to the magnitude of the large row in this group. Finally, we approximate the variance $\left\| (\mathbf{S}^{(l)})^{-1} \right\|_F^2 \approx \sum_{i=1}^{G} M_i^2 / \left[ (N - G)\frac{M_i}{\sum_{i=1}^{G} M_i} \right]$ and select the best $G$ with minimal variance.

3. Use the grouping of rows described in Step 2 to construct the block Householder quantizer.

# E   Experimental Setup

**Model:**   Our ResNet56-v2 model for CIFAR10 directly follows the original paper [40]. For the ResNet18/50 model, we adopt a slightly modified version, ResNetv1.5 [45]. The difference between v1.5 and v1 is, in the bottleneck blocks which requires downsampling, v1 has stride = 2 in the first 1x1 convolution, whereas v1.5 has stride = 2 in the 3x3 convolution. According to the authors, this difference makes v1.5 slightly more accurate ($\sim$0.5%) than v1, but comes with a small performance drawback ($\sim$5% images-per-second).

**Model hyperparameter:**   For CIFAR10, we follow the hyperparameter settings from the original papers [29, 40], with weight decay of $10^{-4}$.

For ImageNet, we keep all hyperparameters unchanged from [45], which has label smoothing=0.1, and weight decay=1/32768.

**Optimizer hyperparameter:**   For CIFAR10, we follow the original paper [29], with a batch size of 128, initial learning rate of 0.1, and momentum 0.9. We train for 200 epochs.

For ImageNet, we follow [45], which has a momentum of 0.875. Due to limited device memory, we set the batch size to 50 per GPU with 8 GPUs in total, the initial learning rate is 0.4. We train for 90 epochs, and the first 4 epochs has linear warmup of the learning rate.

For both datasets, we use a cosine learning rate schedule, following [45].

**Quantization:**   We follow the settings in [20]. All the linear layers are quantized, where the forward propagation is

$$\mathbf{F}^{(l)}\left( \tilde{\mathbf{H}}^{(l-1)}; \tilde{\mathbf{\Theta}}^{(l)} \right) = \tilde{\mathbf{H}}^{(l-1)}\tilde{\mathbf{\Theta}}^{(l)}, \text{where } \tilde{\mathbf{H}}^{(l-1)} = Q_f\left( \mathbf{H}^{(l-1)} \right), \ \tilde{\mathbf{\Theta}}^{(l)} = Q_\theta\left( \mathbf{\Theta}^{(l)} \right),$$

both $Q_f(\cdot)$ and $Q_\theta(\cdot)$ are deterministic PTQs that quantizes to 8-bit. The back propagation is

$$\hat{\nabla}_{\mathbf{\Theta}^{(l)}} = \tilde{\mathbf{H}}^{(l-1)^\top} Q_{b1}(\hat{\nabla}_{\mathbf{H}^{(l)}}), \quad \hat{\nabla}_{\mathbf{H}^{(l-1)}} = Q_{b2}(\hat{\nabla}_{\mathbf{H}^{(l)}})\tilde{\mathbf{\Theta}}^{(l)^\top},$$

with gradient bifurcation [20]. We set $Q_{b1}$ to a 8-bit stochastic PTQ, and $Q_{b2}$ to PTQ, PSQ, or BHQ with 4-8 bits. The original paper [20] set $Q_{b1}$ as an identity mapping (i.e., not quantized), and $Q_{b2}$ to be 8-bit stochastic PTQ.

We quantize the inputs and gradients of batch normalization layers, as described in our framework.

**Number of training / evaluation runs:**   Due to the limited amount of computation resources, we train on each setting for only once.

**Runtime & Computing Infrastructure:**   Following [20], we simulate the training with FP32. Our simulator runs approximately 3 times slower than FP32 counterparts. We utilize a machine with 8 RTX 2080Ti GPUs for training.

# F   Additional Experimental Results

Figure 6: CIFAR10 convergence curves.

Figure 7: ResNet18 on ImageNet convergence curves.

Figure 8: ResNet50 convergence curves.