[Reviews · NeurIPS 2020]

Review 1

Summary and Contributions: The paper introduces a new method for fully-quantized training (FQT) of neural networks (where activations, weights and gradients are all quantized). This method stems from a statistical estimation of a quantization aware training (QAT) wherein gradients are kept in full precision. The authors analyze the variance increase of gradients in FQT using a per-tensor quantizer and show introduce two novel quantizers that can provably lower the variance. The authors show empirically that these methods produce better quantization (using more of the available data bins) and perform favorably in image classification benchmarks. **** Post rebuttal **** Authors mostly addressed my concerns by adding more results and rough computational cost analysis. I will raise my score and will be happy to see this work at Neurips.

Strengths: The authors have done a good job at motivating their approach, while providing a thorough background on different quantization training schemes. Their suggested methods are shown to provide benefits both in theory (lower quantization noise) as well as empirical (histogram and model accuracy). Their approach is novel (to best of my knowledge) and seem well devised enough to have a significant contribution to this field and the NeurIPS community.

Weaknesses: Empirical evaluation is also a bit weak in my opinion: - No baseline is used in plots or tables (full precision training, not just QAT) - Results are just for image classification tasks and for a few models (two). I think that authors should add at least another domain (translation, language-modeling, etc) and another model (transformers, rnns, densenets, etc) - There is no discussion on computational cost of their models (except of an O complexity mention in terms of FP32 ops). Specifically, how much additional resources are required for the intoduced quantization schemes? do they require additional considerations for future possible hardware (e.g, sorting required for the Block House Holder)

Correctness: Theoretical analysis seems well established (although I didn't verify proofs available in appendix). Empirical methodology is decent, although lacking in scope (see Weakness part).

Clarity: Writing quality can be improved in my view, as I feel that the background section (section 2) is too obfuscated, white unnecessary notational burden (for example, using Jacobians to represent the gradients). Instead, I suggest authors to shorten this section and use the additional space to better explain technical details of their methods. For example, I feel the Block Householder section (4.2) can benefit from a clear use case example.

Relation to Prior Work: The work clearly relates to previous work and explains the differences and benefits of the authors' approach.

Reproducibility: Yes

Additional Feedback: I consider raising my score if additional experiments are added and computational overhead is discussed (as described in Weakness section)


Review 2

Summary and Contributions: This paper introduces a novel framework for accelerating the training phase of deep neural networks. In light of the introduced framework, the authors show that fully quantized training (FQT) method estimates an unbiased gradient and provide a formula for analyzing the variance of the FQT gradient. In addition, two novel methods are proposed to quantize the gradient and extensive experimental results demonstrate the effectiveness.

Strengths: + This paper provides a theoretical understanding of FQT and shows that gradient used in FQT is actually an unbiased estimator of the QAT gradient. + The paper also shows how to devise methods for reducing the variance in the FQT, with solid and promising experiment results. + In-depth analyses (e.g. Fig3 (a)) are provided, which is a strong support for the introduced framework.

Weaknesses: Although this is a decent paper, I have a concern. - The proposed methods (PSQ, BHQ) seem more complicated than PTQ, so these may lead to higher computing cost while FQT aims to reduce the time cost. Thus, it is necessary to report the time cost of different methods. I would like to see a section to discuss the results in Table 1. Specifically, the performance gain on ResNet50 is larger than that on ResNet18, which may imply that the proposed methods have limited effect on relatively small models. So, how to understand the phenomenon?

Correctness: Yes

Clarity: Yes

Relation to Prior Work: Yes

Reproducibility: Yes

Additional Feedback: Please address my detailed comments mentioned in ‘Weaknesses’.


Review 3

Summary and Contributions: The authors analyze the effect of gradient quantization for quantized training in a principled fashion, and introduce two methods that reduce the variance of the gradients when doing quantized training. *** Post Rebuttal *** It's great that the authors added a computational overhead result! Still I hold that if FQT is compared to QAT, you should quantize the weights and not keep shadow weights. This is what I meant with having the actual weights quantized, and the updates quantized as well. In most FQT applications that are parallelized in compute, you are very often memory movement bound, meaning you're playing a game of reducing memory as much as possible. The gradients are calculated on the fly, used and discarded in the backward pass, the memory overhead of them is small. I would have liked to see a similar analysis for the actual weight update that's quantized in this paper. I believe the authors missed my point on the computing the ranges on the fly. In paragraph 4.1, the per-sample quantizer is explained. In goes the activation gradient matrix, and S is calculated based on R(grad of activation matrix) = max - min of the grad of activation matrix. Now here's the catch22, if you want to calculate the min and max of the grad of activation matrix, you need to calculate the grad of activation matrix. On any device that's not a super efficiently cached CPU, this means you have to write the output of your 8x8 bit multiplication to TCM or DDR, then calculate the min and max, and THEN requantize the grad of the activation matrix to 8 bit again. This means you're writing a 16 bit tensor (or however big your accumulator is) to memory, and I reckon that for nearly all efficient hardware this is a bottleneck in terms of power and latency (since with FQT, you're often memory restricted). I hope I didn't misinterpret this; but I checked the paragraph several times to make sure this is according to the authors intended method. I stand corrected on the bias and variance! Thanks for explaining. Updated the score to a 7 from 6, because the authors partially answered my worries/critique.

Strengths: The paper is very well-written, and despite parts being complex, easy to follow and read. The theoretical analysis is sound and interesting, and it sheds a good light on what's actually going on with quantized training when looking at the gradients. Showing the relationships in the variance and the bit-widths and how these terms interact with the general training variance was insightful to me. Of the two methods introduced to reduce the variance of quantized training, one of the methods seems useful to me and is something I reckon many will explore afterwards. I have also not seen this approach before, although the fact that the gradients have per-example different quantization-ranges is something that I've seen before. In the experimental section, the results where the methods show reduced variance are really good. And despite the experimental results being OK and not very comprehensive, the fact that the authors show that there is actually reduced variance is good enough for me.

Weaknesses: The paper has some weaknesses that I reckon when addressed, make the paper even stronger. - There is no discussion on quantization of the weights and the weight update. The single largest bottleneck in quantized training is not the actual computation, but the memory movement. For this reason, the weights should be quantized as well. But the paper does not analyze what happens when the weights and the weight update are quantized, only what happens to the quantized gradients. An extra analysis on this would make the paper a lot stronger, and a lot more practical - I got excited when reading that the paper would adopt a statistical approach, rather than analyzing the worst-case behavior of the gradient. However, I don't see this come to full fruition. In the core Theorem 2 the major approximation is taking the largest eigenvalue of the compounded gradient update gamma. I wouldn't consider this to be a very tight bound, and just as much a worst-case scenario as any other method that analyzes how errors propagate over the network. I wonder how tight this bound is; perhaps the authors could consider to calculate this value and compare it empirically. That would make the paper a lot stronger. - The per-sample quantizer makes sense. It is very reminiscent of doing per-channel quantization on the weights. It might be good to reference the relationship here. There is also a definite drawback to this method, in that you need to do extra 'floating-point operations' on the accumulator. Not all hardware will nicely support this; and if it does not natively, you would have to write the full 32-bit tensor to working memory to process the activation quantization as it's suggested here. That would forego almost all gains from doing the gradient quantization in the first place. The pure computing part of the PSQ method is likely not the biggest issue, as it's just multiplication with a few scaling factors, which can be done in the accumulator. However, finding the range R (max(grad) - min(grad)) on the fly is cumbersome. In these calculations on any vector/tensor processor like a DSP or NPU, you will not have calculated the full output for an example before you have to write your accumulator contents to TCM or DDR. Thus you will not know the actual range. The only way to make this process efficient is to calculate the ranges before doing the computation. You could do this e.g. based on the cross-entropy loss at the end of the network. Some discussion on this would be good to make the methods in this paper actually useful in practice. - The Block Householder Quantizer seems sensible from a theoretical perspective, but following the reasoning above, is infeasible in practice. You would need dedicated silicon on the accumulators to do this processing, which is non-trivial and probably expensive. The gains from the results section also wouldn't warrant the inclusion of this method on a chip. And again, if you don't dedicate sillicon the the processing in the accumulator, you will likely have to write your entire feature-map to DDR/TCM in 32 bits, foregoing all gains you would get from activation quantization. Thus I don't think the Block Householder method is actually useful in practice.

Correctness: I think the methodologies are correct. The theoretical analysis seems sound; the experimental section is not great nor exhaustive, but since the authors show the gradients decrease and back-up their theoretical evaluations with the useful plots in fig 4; the experimental results on networks would show the aptitude of the authors to train a network more than that they would constitute a proof for the actual method.

Clarity: Very well written, easy to follow, good grammar, didn't spot a single mistake.

Relation to Prior Work: Yes. The authors show a clear understanding of the literature in their related work section, and describe very clearly how their method differs from other existing methods.

Reproducibility: Yes

Additional Feedback: If the comments in the weakness session are addressed, I'm willing to increase my rating.

[Author Response · NeurIPS 2020]

We thank all the reviewers for acknowledging our novel contributions and providing valuable feedback. Below,
we address the common concerns followed by detailed comments from each reviewer. We will include additional
experimental results and discussions, and revise the final version according to reviewers' suggestions.

**From R1, R2, and R3: Discussion of the computational cost.** The actual time cost is highly platform-specific. A
complete hardware-algorithm codesign is out of the scope of this paper, which mostly focuses on the theoretical
properties of gradient quantization. Nevertheless, as requested by the reviewers, we investigate the quantization
overhead for a $(N = 128, C = 64, H = W = 56)$ convolutional layer on a single CPU core. In this case, the actual
convolution takes 480ms. Finding the range takes 11ms for PTQ and 24ms for PSQ and BHQ. Additionally, it takes
BHQ 3μs for finding the optimal transformation (including sorting), and 21ms to perform the transformation with
sparse-dense matrix multiplication. Therefore, the overhead for all the quantizers is small relative to the convolution.

As also pointed out by R1 and R3, dedicated hardware implementations may involve more subtleties. Instead of
per-sample FP32 ranges, hardware may favor a per-tensor range with per-sample shift values ([0, 4] is enough according
to our experience), where the accumulators are shifted before the final summation. We leave this as future work.

**From R1 and R2: Evaluation in multiple domains.** Upon reviewers' request, we test FQT on a transformer for
machine translation on the IWSLT14' En-DE dataset. We quantize all the dense layers in the model and apply per-token
quantization for PSQ and BHQ. Following the settings in the original paper, we fix the activation and weight to 8-bit,
and vary the gradient bitwidth. The validation BLEU score, as well as the gradient variance, are shown in the table and
figure below. The conclusion is the same as with the original paper, where BHQ achieves 3 fewer bits than PTQ.

**From R1: No full-precision baseline is used in plots or tables.** There could be some misunderstanding. We already
reported the full-precision results as "Exact" in all our experiments (in both Figure 3 and Table 1).

**From R2: Discuss the results in Table 1.** We will discuss in the final version. Specially, we believe the improvement
is larger on ResNet50 because its training accuracy reaches to 100% more quickly, so the gradient is sparser.

**From R3: The paper does not analyze weight quantization.** In our framework, the weight is indeed quantized in
the sense that only the quantized weight is required for forward and backward computation (Fig. 1). However, our main
focus is on studying the gap between QAT and FQT, i.e., the impact of gradient quantization. Therefore, the analysis of
weight quantization, which is a special case of QAT, is out of the scope of this paper. Such analysis can be found in any
QAT theory papers (e.g., arXiv 1903.05662), and is compatible with the gradient quantization analysis in this paper.

**From R3: Theorem 2 approximates with the largest eigenvalue, which is
still a worst-case bound.** By taking the statistical approach, we study the exact
*bias and variance* of the gradient, rather than the worst-case distance or angle.
In this way, we can prove the unbiased gradient (Theorem 1) as well as an **exact**
formula of the variance (Eq. 7). These results are not available in the worst-case
analysis [20, 22]. The upper bound Eq. 8 mentioned by the reviewer still takes
a statistical approach, since it upper bounds the *variance*, not the *distance*. Eq. 8
is more useful for intuitive explanations and designing better quantizers (PTQ
and PSQ). Even with this imperfect bound we derive useful quantizers, and it is
possible to design even better quantizers (e.g., mixed precision) based on more
precise bounds of Eq. 7, where we leave as future work.

**Table:** Validation BLEU score on the machine translation task.

| Alg. | PTQ | PSQ | BHQ |
|------|------|------|------|
| Exact | 34.55 | – | – |
| QAT | 34.47 | – | – |
| 8-bit | 34.33 | 34.39 | **34.51** |
| 5-bit | 0.02 | 33.17 | **33.70** |

More deeply, we can directly analyze Eq. 7, where each term is the impact of
the $l$-th layer quantizer to the $k$-th layer weight. We compute all the $O(L^2)$
terms and find that quantizers only impact nearby layers, and the impact decays
exponentially with $l - k$. In this case, a lower bound consists only the $k = l$
terms is still reasonably accurate. Each term of this lower bound is proportional
to the terms in Eq. 8, up to a constant irrelevant with the quantizer parameters.

**From R3: Computing the range on the fly is cumbersome.** We suspect there
is some misunderstanding. Our PSQ quantizes the operands before the actual
low-bitwidth multiplication. The accumulator value is known at this point since
it is computed by the last layer's matrix multiplication, which is already finished.

**Figure:** Gradient variance on the machine translation task.

[Meta-Review · NeurIPS 2020]

Three knowledgeable reviewers were all positive, and agreed that this is a novel and well motivated work, which has benefits both in theory (lower quantization noise) as well as empirical (histogram and model accuracy)